# *Trichoderma harzianum* Inoculation Reduces the Incidence of Clubroot Disease in Chinese Cabbage by Regulating the Rhizosphere Microbial Community

**DOI:** 10.3390/microorganisms8091325

**Published:** 2020-08-31

**Authors:** Junhui Li, Joshua Philp, Jishun Li, Yanli Wei, Hongmei Li, Kai Yang, Maarten Ryder, Ruey Toh, Yi Zhou, Matthew D. Denton, Jindong Hu, Yan Wang

**Affiliations:** 1School of Bioengineering, Qilu University of Technology (Shandong Academy of Sciences), Jinan 250353, China; rancho_lee@yeah.net; 2Shandong Provincial Key Laboratory of Applied Microbiology, Ecology Institute, Qilu University of Technology (Shandong Academy of Sciences), Jinan 250013, China; yewu2@sdas.org (J.L.); yanli_wei@163.com (Y.W.); hmlihm@163.com (H.L.); menshenlai@163.com (K.Y.); 3China-Australia Joint Laboratory for Soil Ecological Health and Remediation, Ecology Institute, Qilu University of Technology (Shandong Academy of Sciences), Jinan 250013, China; joshua.philp@adelaide.edu.au (J.P.); maarten.ryder@adelaide.edu.au (M.R.); ruey.toh@adelaide.edu.au (R.T.); yi.zhou@adelaide.edu.au (Y.Z.); matthew.denton@adelaide.edu.au (M.D.D.); 4School of Agriculture, Food and Wine, The University of Adelaide, Urrbrae 5064, Australia

**Keywords:** clubroot, high-throughput amplicon, *Trichoderma harzianum*, *Plasmodiaphora brassicae*, microbial community

## Abstract

Clubroot is a disease of cruciferous crops that causes significant economic losses to vegetable production worldwide. We applied high-throughput amplicon sequencing technology to quantify the effect of *Trichoderma*
*harzianum* LTR-2 inoculation on the rhizosphere community of Chinese cabbage (*Brassica rapa* subsp. *pekinensis* cv. Jiaozhou) in a commercial production area. *T. harzianum* inoculation of cabbage reduced the incidence of clubroot disease by 45.4% (*p* < 0.05). The disease control efficacy (PDIDS) was 63%. This reduction in disease incidence and severity coincided with a drastic reduction in both the relative abundance of *Plasmodiaphora brassicae*, the causative pathogen of cabbage clubroot disease, and its copy number in rhizosphere soil. Pathogenic fungi *Alternaria* and *Fusarium* were also negatively associated with *Trichoderma* inoculation according to co-occurrence network analysis. Inoculation drastically reduced the relative abundance of the dominant bacterial genera *Delftia* and *Pseudomonas*, whilst increasing others including *Bacillus*. Our results demonstrate that *T. harzianum* LTR-2 is an effective biological control agent for cabbage clubroot, which acts through modulation of the soil and rhizosphere microbial community.

## 1. Introduction

Clubroot is an economically significant disease of cruciferous crops caused by the biotrophic pathogen *Plasmodiophora brassicae* Woronin [1,2,3]. Root cells infected by *P*. *brassicae* undergo abnormal cell division and enlargement, resulting in the formation of spindle-like, spherical, knobby or club-shaped root galls [2,4]. *P*. *brassicae* consumes carbohydrates from the viable cells in galls, causing the galls to become sinks for nutrients [5,6]. This significantly alters the morphology, development, and physiology of the diseased plants [7], and leads to substantial yield loss [3,8,9].

Current methods to control *P. brassicae* include crop rotation [10,11], liming [12], and agricultural chemicals such as quintozene, chlorothalonil, flusulfamide and fluazinam [13,14,15]. Although chemical methods can be effective, they have drawbacks associated with fungicide resistance, chemical pollution and impacts on food quality [16]. Accordingly, biological control methods are being increasingly investigated as a more viable alternative for controlling *P. brassicae* [15,17].

*Trichoderma* is one of the most widespread biological agents currently used in agriculture to control diseases [18,19,20,21,22,23,24], being present in more than 60% of registered biological pesticides worldwide [25,26,27]. *T. harzianum* [28,29] was found to have a controlling effect on clubroot disease of Chinese cabbage in pot experiments [23]. The use of biocides usually affects the microbial community including bacteria and fungi [30,31]; however, little detail is available about the effect of *T. harzianum* on plant roots and microbial communities in the rhizosphere. It has been reported from pot experiments and subsequent non-metric multidimensional scaling tests that *T. harzianum* alters the rhizosphere bacterial community [32,33,34]; however, its interactions with *P. brassicae* and with the rhizosphere fungal community remain unquantified.

With the development of high-throughput sequencing technology, it has become possible to explore dynamic changes in microbial communities with the application of biological control agents [15,33,34,35,36]. Furthermore, the use of quantitative polymerase chain reaction (qPCR) based technologies enables quantification of microorganisms in soil [8,37,38,39,40]. These methods were used in this study to facilitate the assessment of *Trichoderma* biological control and its impact upon the *P. brassicae* population. This paper aims to quantify (1) the effect of *Trichoderma* inoculation in the control of *P. brassicae* clubroot disease, and (2) the additional impact of *T*. *harzianum* LTR-2 on rhizosphere microbial communities.

## 2. Materials and Methods

### 2.1. Research Site

Research was conducted in Jiaozhou, Shandong Province, China (36°0′ N, 119°8′ E) during August–November 2019. Jiaozhou is a renowned Chinese cabbage (*Brassica rapa* subsp. *pekinensis* cv. Jiaozhou) production area; however, the yield of cabbage has declined sharply in recent years due to clubroot disease. A trial site was established in a field known to produce cabbage with a high incidence and severity of clubroot. The field was mechanically ploughed to 20 cm depth and then divided into four plots (3 × 30 m^2^) separated by 1.5 m gaps. Cabbages were sown with a between-row spacing of 0.6 m and a within-row spacing of 0.35 m.

### 2.2. Experimental Design

The experiment was based on a two-factor factorial design with three replications. Factors included +/− bioformulation (i.e., *T. harzianum* LTR-2 and control) and sampling in two microsite compartments (i.e., bulk soil and rhizosphere soil), which generated four combinations of BsT (bulk soil with *T. harzianum*), BsC (bulk soil control), RsT (rhizosphere soil with *T. harzianum*) and RsC (rhizosphere soil control).

### 2.3. Bioformulation

The bioformulation was a freshly prepared spore suspension (10^8^ spores/mL) of laboratory-sourced *T*. *harzianum* strain LTR-2 (5 × 10^10^ spores/g) with distilled water. For the treated plots, seeds were soaked in bioformulation for 10 min prior to sowing and a freshly made mixture of bioformulation and sterilized sand (ratio 1:5) was then sprinkled onto the plots at 10 kg/ha.

### 2.4. Sample Collection

After 3 months of growth, 30 root systems were extracted from under randomly selected cabbages in each plot. Soil 20 to 30 mm from the roots was defined as bulk soil and was transferred into sterile sample bags using sterile brushes. Soil that adhered to roots up to 1 mm from the root surface was rinsed using sterile water, and centrifuged to produce samples of rhizosphere soil. For each plot, the collected rhizosphere soil and bulk soil samples were separately combined and stored at –80 °C prior to analysis.

### 2.5. Assessment of Clubroot Disease Incidence and Severity

Incidence of clubroot disease in cabbages under each inoculation treatment was assessed relative to bioformulation treatments using a bioassay of disease incidence from a random sample of 150 cabbages with intact root systems from each plot. Disease severity of each cabbage was graded on a five-level scale [15], defined as: 0 = no galling; 1 = fibrous root swelling, no main root swelling; 2 = main root swollen with diameter 2 to 3 times the stem base; 3 = main root swollen with diameter 3 to 4 times the stem base, 4 = main root swollen with a diameter greater than 4 times the stem base.

Disease index and control effect were calculated as follows:$$\begin{array}{c} \mathrm{Disease}\;\mathrm{incidence}\;(\%)\;=\;\frac{\left(number\;of\;diseased\;plants\right)}{\left(total\;number\;of\;investigated\;plants\right)}\times 100\\ \mathrm{Disease}\;\mathrm{index}\;(\%)\;=\;\frac{\sum (\left(number\;of\;diseased\;plants\;at\;each\;stage\right)\times (relative\;value))}{\left(total\;number\;of\;plants\;under\;investigation\right)\;\times \left(highest\;incidence\;of\;disease\right)}\times 100\\ \mathrm{Control}\;\mathrm{effect}\;(\mathrm{PDIDS})\;(\%)\;=\;\frac{\left(\mathrm{disease}\;\mathrm{index}\;\mathrm{control}\;\mathrm{group}\right)\;-\left(\mathrm{disease}\;\mathrm{index}\;\mathrm{treated}\;\mathrm{group}\right)}{\left(\mathrm{disease}\;\mathrm{index}\;\mathrm{control}\;\mathrm{group}\right)}\times 100 \end{array}$$

### 2.6. Bacterial and Fungal Genomic DNA Extraction

Total DNA was extracted from 0.5 g samples of the bulk soil and rhizosphere soil using the soil genomic DNA rapid extraction kit (Bio-Bioengineering Co., Ltd., Shanghai, China). The extracted DNA was electrophoresed on a 1% agarose gel. Concentrations were measured by a BioSpec-nano Ultra Micro Spectrophotometer (Shimadzu, Kyoto, Japan).

### 2.7. PCR Amplification and High-Throughput Sequencing

The extracted soil genomic samples were sequenced amplicons using Illumina-MiSeq. The specific method was as follows:

Both regions of the 16S rDNA and ITS gene were separately amplified using a nested polymerase chain reaction (PCR) approach. In the first amplification reaction, primer 341F-805R and primer ITS3-ITS4 (see Table 1) were used for 16S rDNA and ITS rDNA amplification, respectively, at 0.2 μmol/L primer. This reaction was prepared in a final volume of 30 uL, containing 15 μL of 2 × Taq master Mix (Thermo, New York, NY, USA), 1 μL each of primers (10 μmol/L), and 20 ng of template DNA. Amplification conditions were: 94 °C for 3 min, 5 cycles of amplification of 94 °C for 30 s, 45 °C for 20 s, 65 °C for 30 s, and 20 cycles of amplification of 94 °C for 20 s, 55 °C for 20 s, 72 °C for 30 s, and 20 cycles of amplification, followed by extension at 72 °C for 300 s. The second round of amplification used Illumina bridge PCR compatible primers, with the first round of PCR products as templates. The reaction system was the same as above. Amplification conditions were: 95 °C for 30 s, 95 °C for 15 s, 55 °C for 15 s, 72 °C for 30 s, 5 cycles of amplification, and 72 °C extension for 300 s. PCR products were purified and recovered, and then the Qubit 3.0 DNA detection kit was used to accurately quantify the recovered DNA, so as to facilitate Illumina-MiSeq sequencing at an equal ratio of 1:1.

### 2.8. qPCR Amplification

The primers were adopted from Li [41]: PBF (5’ GAACGGGTTCACAGACTAG-AT-3’); PBR (5’-GCCCACTGTGTTAATGATCC-3’). The amplified fragment length was 200 bp, synthesized by Changsheng Biotechnology Co., Ltd. (Dingguo, Beijing, China). The amplified fragments of the primers were ligated with the pTG-T-vector Fast Ligation Kit (LMAI Bio, Shanghai, China) to construct a standard plasmid.

Using the SYBR Green I fluorescent dye method, the qPCR reaction system (20 μL) contained 1 × SYBR Premix Ex TaqTM 10 μL, primer PbF/PbR (0.2 μmol/L) 0.4 μL each; DNA template 2 μL; dd H_2_O (Sterilization) 7.2 μL. Water was used instead of DNA in the blank control. The qPCR reaction procedure was performed using an iCycler iQ5 (Bio-Rad, California, USA) real-time PCR instrument. Pre-denaturation was at 95 °C for 3 min, denaturation was at 95 °C for 10 s, annealing was at 56 °C for 30 s, extension was at 72 °C for 20 s, for 40 cycles. Fluorescence was acquired multiple times in the extended phase of each cycle (72 °C). The dissolution profile was analyzed after PCR amplification to verify the specificity of the amplification. The procedure for the dissolution curve was: 95 °C, 1 min; 56 ° C, 1 min; from 56 °C for every 0.5 °C for 10 s, and then continuous increase 89 times (to 95 °C).

### 2.9. Processing of Bioinformatics and Data Analysis

The original data was uploaded to the GSA (Genome Sequence Archive) database to submit and save the original information for sequencing (Login number: CRA002534). The original data were sequenced by removing primer adapter sequences, removing low-quality bases (Phred Quality Score = 20), and splicing, discarding sequences shorter than 100 bp, and removing the specific amplification sequences and chimeras to obtain the effective value of each sample’s sequence data. The 16S and 18S sequences were divided into operational taxonomic units (OTUs) with 97% as the threshold. Annotation of the representative sequences of the OTUs was done using the Mothur [42] method. SILVA [43] and Unite [36] databases were used to perform species annotation analysis, obtain taxonomic information and calculate the community composition of each sample at the phylum, class, order, family, genus, and species classification levels. Subsequent analysis first sorted the data using the longest sequence in each OTU as a representative sequence, looking for homologous sequences in the NCBI database through BLAST, and identifying the species with the greatest similarity and more than 95% confidence sequence.

#### 2.9.1. Differential Analysis of Bacteria and Fungi

R’s vegan software package was used to calculate the alpha diversity index (Shannon, Simpson index, Chao1 and ACE and Observed Richness of OTUs). Analysis of similarity between microbial communities was conducted by principal coordinate analysis (PCoA) according to weighted unifrac distance matrix between sample OTU compositions. R’s ggplot2 software package was used to plot data.

#### 2.9.2. Differential Analysis of Fungi

LEfSe [44] (http://huttenhower.sph.harvard.edu/galaxy/) online analysis software was used to analyze differences between the microbiome. OmicShare (https://www.omicshare.com) online analysis software was used to analyze the relationship between samples and species and to generate a circos [45] species relationship map. The FUNGuild [46] database was used to identify fungal OTUs belonging to phytopathogenic fungi and to calculate their relative abundance. MUNA [47] (http://ieg4.rccc.ou.edu/mena) online analysis software was used to analyze the number of OTUs related to predicted plant pathogens in the rhizosphere soil and control soil, which were then imported into Cytoscape (3.6.1) to generate co-occurrence networks.

## 3. Results

### 3.1. Field Test Results

*T*. *harzianum* LTR-2 effectively controlled clubroot of Chinese cabbage in the field (Table 2). On the basis of the incidence rate of 96.7%, the application of *Trichoderma* reduced the incidence of clubroot to 51.3%. The disease index was significantly lower in the inoculated group compared with the control group (23.2% and 62.5%). The disease control efficacy (PDIDS) was 63%.

### 3.2. Alpha Diversity Analysis

The Shannon and Simpson indices of bacteria in *Trichoderma*-treated rhizosphere and bulk soil were higher than those of the corresponding untreated rhizosphere and bulk soil (Figure 1A,B). The Shannon and Simpson indices of fungi were slightly lower in the treated bulk and rhizosphere soil relative to the untreated bulk and rhizosphere soil, respectively. Within treatments, bulk soil had higher Shannon and Simpson indices than rhizosphere soil for both bacteria and fungi. Treated rhizosphere soil had significantly more bacterial and fungal OTUs than untreated rhizosphere soil, whereas treated bulk soil had less bacterial and more fungal OTUs than untreated bulk soil (Figure 1C).

From Figure 1D, the ACE index of bacteria and fungi in the treated rhizosphere soil was significantly higher than that of the untreated rhizosphere soil. There were no significant differences in ACE index for bacteria and fungi between treatments for bulk soil.

### 3.3. Beta Diversity Analysis

#### 3.3.1. Principal Component Analysis

UniFrac-weighted PCA based on the composition of bacterial and fungal OTUs are shown in Figure 2. The distance between the fungal OTU communities in rhizosphere soil and bulk soil in the control was much greater than in the treatment, indicating that bioformulation reduced the variation in OTU composition between rhizosphere and bulk soil. There was no significant difference among samples in the bacterial community.

#### 3.3.2. Linear Discriminant Analysis Effect Size (LEfSe)

LEfSe analysis identified significant differences in the abundance of different genera between bioformulation-treated and control rhizosphere soil (Figure 3). In control soil, *P. brassicae* was significantly more abundant, as was *Dioszegia, Mucor.* and *Devriesia.* Within the treated soil, significantly more abundant genera were *Trichoderma, Zopfiella, Orbiliaceae, Gloeoporus, Chytridiaceae, Talaromyces, Mortierella* and *Oisiodendron.*

### 3.4. Community Analysis

#### 3.4.1. Bacteria

The structure of bacterial populations in the rhizosphere soil and the bulk soil was similar at the phylum level, and there was no obvious change between treatments (Figure 4A).

*Pseudomonas* had the highest relative abundance in untreated rhizosphere and bulk soil (Figure 4B). *Delftia* had a high relative abundance similar to *Pseudomonas* in untreated rhizosphere soil, whereas *Bacillus*, *Delftia*, *Massilia* and *Sphingomonas* were similarly abundant to each other in untreated bulk soil.

Treatment with *T. harzianum* significantly changed the relative abundance of several genera in the rhizosphere and bulk soil. The relative abundance of *Delftia* in rhizosphere and bulk soil was much lower when treated with *T. harzianum,* being 29.2% and 5.3% in untreated rhizosphere and bulk soil, respectively, and 0.3% and 0.1% in treated rhizosphere and bulk soil. Similarly, the relative abundance of *Pseudomonas* was much lower in both soil types when treated with *T. harzianum,* being respectively 29.2% and 19.1% in untreated rhizosphere and bulk soil, respectively, and 8.5% and 3.1% in treated rhizosphere and bulk soil. The major reduction of *Delftia* and *Pseudomonas* in the rhizosphere after treatment with *T. harzianum* was accompanied by increases in the relative abundance of *Bacillus*, *Massilia*, *Mucilaginibacter*, *Sphingomonas* and others, resulting in a more even distribution of relative abundance between genera.

#### 3.4.2. Fungi

Ascomycota was the fungal phylum with the highest relative abundance in each treatment (Figure 4C). The proportion of Ascomycota in bulk soil and untreated rhizosphere soil was higher than that in treated rhizosphere soil, whereas the proportion of *Olpidiomycota* was greater in rhizosphere soil regardless of treatment. There was no significant effect of inoculation with *T. harzianum* at the level of phylum.

The dominant genera differed according to treatment and soil type (Figure 4D). The genera with the highest relative abundance in untreated bulk soil were *Fusarium* (12.9 %) and *Alternaria* (12.3%). *Plasmodiophora* had a low relative abundance (1.6%). Conversely, *Plasmodiophora* was the dominant fungi in the untreated rhizosphere soil (31.7%). In *T. harzianum* inoculated soils, the relative abundance of *Plasmodiophora* was dramatically lower, being 0.02% in both soil types. The relative abundance of *Alternaria* was also much lower in inoculated rhizosphere soils, (13.8%) than in untreated rhizosphere soils (1.5%). These decreases were accompanied by notable increases in the relative abundance of *Humicola* and *Mortierella* in the rhizosphere soil. Inoculation caused the inoculant, *Trichoderma,* to have the highest relative abundance in the bulk soil (34.3%), whereas its relative abundance in the inoculated rhizosphere soil was much lower (2.3%).

#### 3.4.3. Flora Function, Prediction and Analysis

The plant pathogens in the soil were mainly *Fusarium, Alternaria, Plasmodiophora* and *Ceratocystis*. The relative abundance of plant pathogens in the rhizosphere and bulk soil was significantly lower when treated with *T. harzianum* (Figure 5A), and *Plasmodiophora* was amongst the pathogenic genera that were less abundant (Figure 5B). 

### 3.5. Co-Occurrence Network Analysis

*T. harzianum* occurrence was positively associated with the occurrence of 8 plant pathogen species and negatively associated with the occurrence of 18 (Figure 6). Among the pathogens negatively associated were *P. brassicae* and various *Fusarium* species.

### 3.6. Analysis by qPCR

It was found by qPCR that the copy number of *P*. *brassicae* was significantly higher in the rhizosphere than in untreated bulk soil, potentially due to host specificity. Treatment with *T*. *harzianum* greatly reduced the copy number of *P*. *brassicae* in the rhizosphere soil (*p* < 0.05) by over 40 times (Figure 7). The copy number of *P*. *brassicae* in treated bulk soil was relatively low, however the difference compared to untreated bulk soil was less pronounced (approximately 3.3 times) as there were fewer copies detected in the untreated bulk soil.

## 4. Discussion

Previous reports of *Trichoderma* applications for prevention and control of clubroot have found differing effects [30,48,49,50]. We posited that differences in control effects between studies may be related to the differences in regions and microbial communities. The results of this study showed that inoculation of *T. harzianum* LTR-2 in cabbage soil with a heavy clubroot burden reduced the incidence rate by 45.4%, and its PDIDS was 33% higher than reported in other studies [48], at 63%. This indicates that the strain *T. harzianum* LTR-2 could prevent and control clubroot under the climatic and soil conditions of Jiaozhou, Shandong Province, China.

### 4.1. Fungal Community

There were large and significant differences between the diversity and OTU composition of fungal communities in untreated and treated soil. Alpha analysis showed that the species richness of the soil fungal community in the treatment group was higher than that in the control group, but the diversity of the species decreased. This shows that the introduction of *T. harzianum* did not directly reduce the number of species, but changed their relative abundance. This was similar to the effect of *Trichoderma* reported by Debode [51] on the microbial community of strawberry rhizosphere fungi. Combined with LEfSE analysis as shown in Figure 3, it was determined that *Talaromyces*, *Chytridiaceae*, and *Mortierella*, etc., were statistically significant biomarkers in the rhizosphere microbial community of inoculated cabbage, in addition to the inoculant *T*. *harzianum* itself. This indicates that the application of *T*. *harzianum* had a significant effect on the microbial community structure. In the absence of the *T. harzianum* treatment, pathogenic *P. brassicae*, was a statistically significant component of the rhizosphere, indicating that the application of *T. harzianum* reduces the influence of pathogenic *P. brassicae*. The relative abundance of pathogenic *P. brassicae* in the rhizosphere community was also reduced. The reduction of pathogens had a positive effect on the reduction of the incidence of cabbage clubroot. Combined with the previous research on *Trichoderma* as biological control [19,52,53,54,55], we posit that the control effect is related to the colonization of *Trichoderma* in the rhizosphere of cabbage and subsequent change of fungal microbial community in the roots.

This research identified significant differences in the fungal community structure between treated and untreated soils. Funguild predictions from the amplicon sequencing results indicate that the relative abundance of plant pathogens was significantly lower in *T*. *harzianum* treated rhizosphere soil samples. A negative correlation between *T. harzianum* and 18 plant pathogens, including *P*. *brassicae*, *Fusarium* and *Alternaria* was found through co-occurrence network analysis (Figure 6), which coincided with a drastic reduction in relative abundance in treated soils compared with untreated soils (Figure 4D). Although *Fusarium* is a common plant pathogen in the rhizosphere [56], we did not observe significant root rot when determining the incidence of plant diseases. In conjunction with earlier reports, many *Fusarium* species are not pathogenic, which may be the case for the *Fusarium* species detected in our analysis [57]. Combined with the results of qPCR, the copy number of *P. brassicae* in the rhizosphere treatment group of Chinese cabbage was also significantly reduced by nearly 100 times compared with the control group.

Together, the results indicate that inoculation with *T. harzianum* prevented the development of clubroot in Chinese cabbages by directly reducing the population of *P*. *brassicae*, and by extension, its relative abundance in the microbial community.

### 4.2. Bacterial Community

There were differences between the diversity and OTU composition of bacterial communities in untreated soil and treated soil. Interestingly the effects of application of *Trichoderma* on the diversity of bacterial microorganisms and species richness in cabbage roots was the inverse of the effects on fungi according to the alpha analysis. Decreased abundance of dominant species and increased diversity indicate that the application of *T. harzianum* makes the soil bacterial microbial community composition of cabbage roots more balanced.

Our result had similarities with a previous study that found that the abundance of *Bacillus* was significantly increased after inoculation with *Trichoderma* [30,58]. It is unknown if this increase is driven by a reduction in the population of other genera or an increase in the population of this genus following inoculation. *Bacillus* is a known biological control agent that can effectively inhibit clubroot disease [59], and the increase in relative abundance may have contributed to lower incidence of disease in this study. The potential for a mutually beneficial interaction of *T. harzianum* and *Bacillus* in the rhizosphere of cabbage warrants further investigation.

The relative abundance of *Delftia* and *Pseudomonas* in the untreated rhizosphere of the control uninoculated plants was extremely high compared with the *T. harzianum* inoculated rhizosphere soil. The cause of the decline may be a direct result of inoculation, via a detrimental interaction with *T. harzianum*, or indirect, arising from other changes in the rhizosphere ecology. A possible mechanism is that inoculation with *T. harzianum* reduced the incidence of clubroot symptoms, thereby reducing the exudation of decayed root material into the rhizosphere and drastically altering the ecological niches available to microorganisms including *Delftia* and *Pseudomonas*. It is also possible that bacteria may also have dependencies on *Plasmodiophora* specifically. The drastic reduction of *Delftia* and *Pseudomonas* was also observed in inoculated bulk soil compared with untreated bulk soil, where the influence of the cabbage root ecosystem is expected to be less pronounced. This may indicate that *T. harzianum* had a direct antagonistic effect on *Delftia* and *Pseudomonas.*

Overall, the composition of the microbial community in the inoculated rhizosphere soil was more evenly balanced relative to the uninoculated rhizosphere soil, which was dominated by fewer genera with very high relative abundance.

## 5. Conclusions

Inoculation of a *P. brassicae*-infested soil with *T. harzianum* greatly reduced the incidence and severity of cabbage clubroot in the field. This reduction in disease coincided with significant changes to the microbial community in the rhizosphere and the bulk soil, most notably a significant reduction of plant pathogens, including *P. brassicae*. In addition, the inoculation of *Trichoderma* has a significant impact on the fungal microbial community in the roots of cabbage. *Trichoderma* has become the main marker instead of *P. brassicae*. However, the inoculation of *Trichoderma* did not have a significant impact on the bacterial microbial communities in the soil of the roots of cabbage. Although there are not significant fluctuations in individual bacterial genera, the overall population abundance becomes more stable. Our results indicate that *T. harzianum* can be an effective biological control agent for cabbage clubroot in field situations.

## Figures and Tables

**Figure 1 microorganisms-08-01325-f001:**
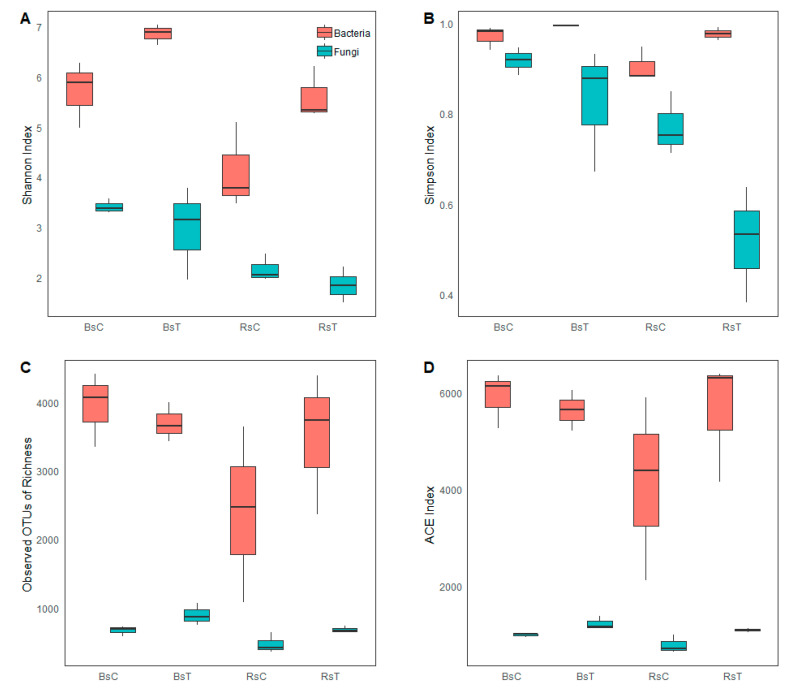
Bacterial and fungal community richness and diversity index of different bioformulation treatments in rhizosphere and bulk soil. (**A**) Shannon index of different treatments of cabbage rhizosphere and bulk soil; (**B**) Simpson index of different treatments of cabbage rhizosphere and bulk soil; (**C**) Observed OTUs of Richness of different treatments of cabbage rhizosphere and bulk soil; (**D**) ACE index of different treatments of cabbage rhizosphere and bulk soil. BsT: Bulk soil with *T. harzianum*, BsC: Bulk soil control; RsT: Rhizosphere soil with *T. harzianum*, and RsC: Rhizosphere soil control.

**Figure 2 microorganisms-08-01325-f002:**
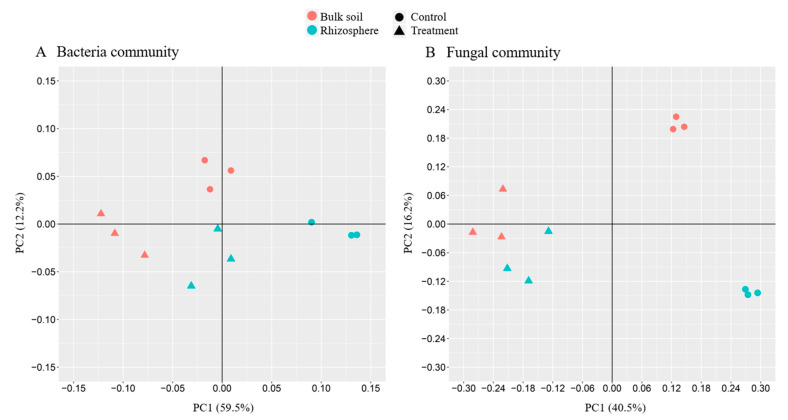
Principal component analysis showing separation between bioformulation and control treatments, in the rhizosphere and bulk soil for bacteria (**A**) and fungi (**B**). The triangle represents the treatment group, the circle represents the control group; the green represents the rhizosphere group, and the red represents the bulk group.

**Figure 3 microorganisms-08-01325-f003:**
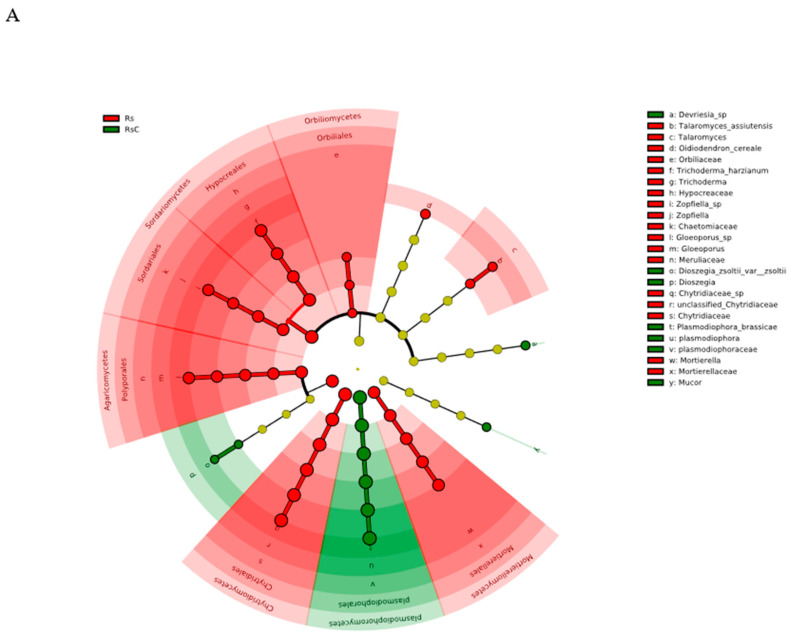
Identification of Chinese cabbage rhizosphere biomarker fungi under different conditions. (**A**) Phylogenetic dendrogram of biomarkers in the Chinese cabbage rhizosphere groups. (**B**) LDA scores of biomarker bacteria for each combination of Cabbage rhizosphere inoculated with *T. harzianum*. (1) Red and green indicate different groups, with the classification of taxa at the level of phylum, class, order, family, and genus shown from the outside to the inside. The red and green nodes in the phylogenetic tree represent significant taxa in the treated and untreated rhizosphere fungal communities, respectively. (2) Yellow nodes represent taxa with no significant difference. Species represented by small yellow nodes were not included. (3) Species with the significant difference that have an LDA score higher than the estimated value. The length of the histogram represents the LDA score; i.e., the degree of influence of taxa with a significant difference between different groups. (LDA score > 3.5, Kruskal–Wallis rank sum test, *p* < 0.05). RsT: Rhizosphere soil with *T. harzianum*. RsC: Rhizosphere soil Control.

**Figure 4 microorganisms-08-01325-f004:**
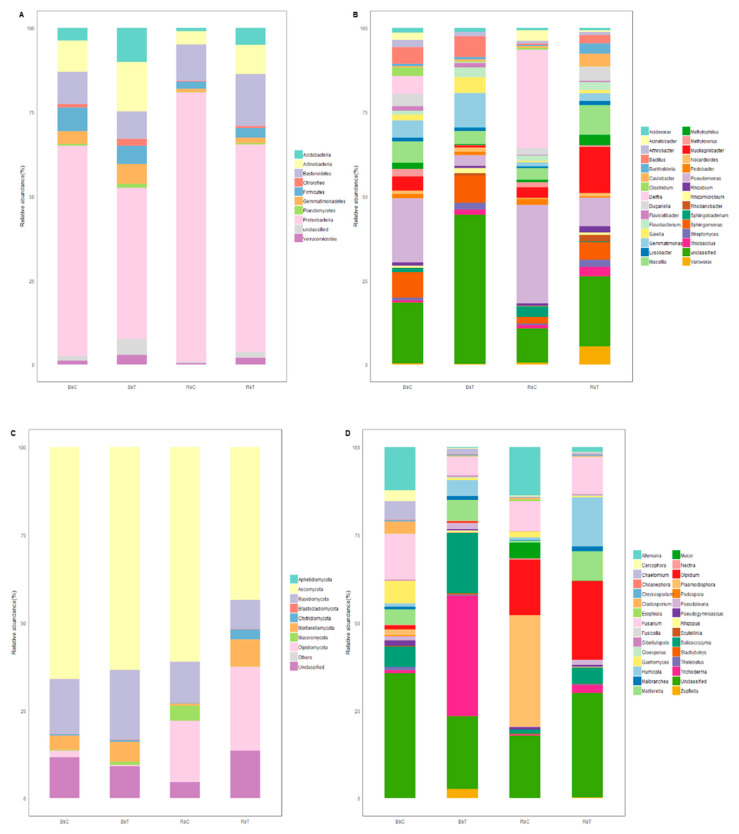
Relative sequence abundance of 10 most abundant phyla and 30 most abundant genera of bacteria and fungi (%) in the four soil samples. (**A**,**B**) are bacteria, (**C**,**D**) are fungi; the data for the top 10 and top 30 taxa were analyzed at phylum and genus level, respectively.

**Figure 5 microorganisms-08-01325-f005:**
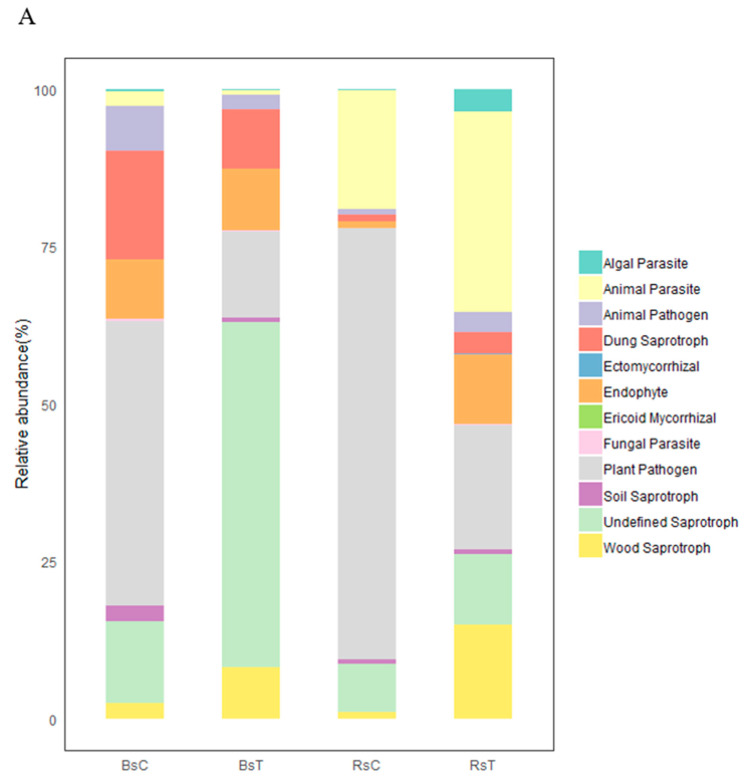
Relative abundance of functional groups within the fungal community under contrasting treatments (**A**) and Circos plot at the genus level for plant pathogens (**B**).

**Figure 6 microorganisms-08-01325-f006:**
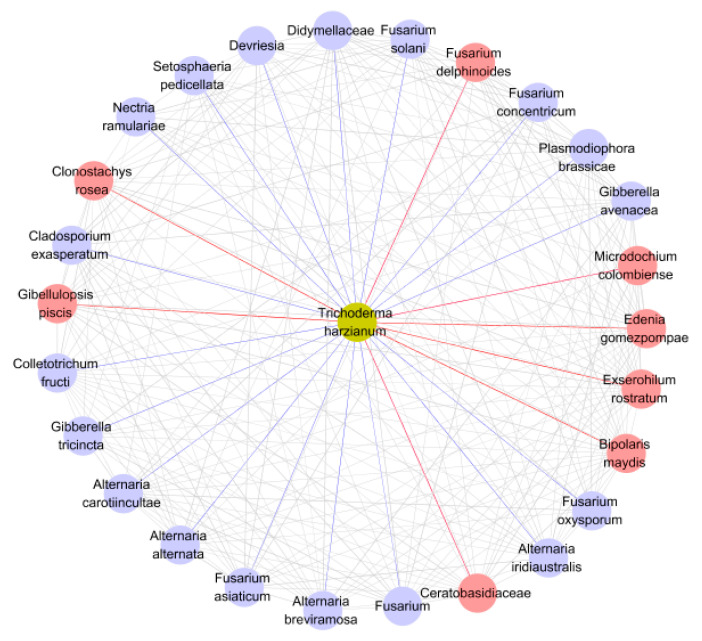
Co-occurrence Network: correlation between plant pathogens and *T*. *harzianum* predicted by Funguild in the rhizosphere fungal community of cabbage. The positive and negative correlations between the plant pathogens associated with *T*. *harzianum* in the environment are marked in red and blue, respectively. *p* < 0.05.

**Figure 7 microorganisms-08-01325-f007:**
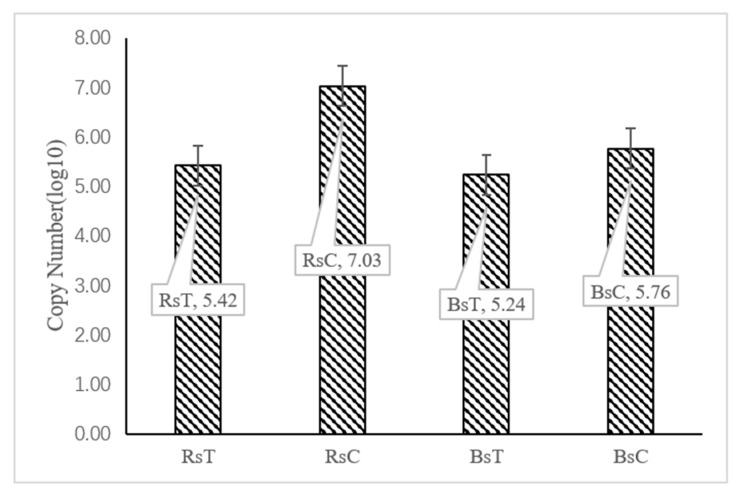
Copy number of *P*. *brassicae* in rhizosphere and bulk soil of different inoculations.

**Table 1 microorganisms-08-01325-t001:** Primers and sequences used in experiments.

Primer	Sequence
341F	CCCTACACGACGCTCTTCCGAT CTGCCTACGGGNGGCWGCAG
805R	GACTGGAGTTCCTTGGCACCCGAGAA TTCCAGACTACHVGGGTATCTAATCC
ITS3	CCCTACACGACGCTCTTCCGATC TNTCCTCCGCTTATTGATATGC
ITS4	GTGACTGGAGTTCCTTGGCACCCGAG AATTCCAGCATCGATGAAGAACGCAGC

**Table 2 microorganisms-08-01325-t002:** Effect of inoculation with *Trichoderma harzianum* LTR-2 on the severity of cabbage clubroot disease *.

	Data	Disease Index	Disease Incidence	PDIDS
Sample				
Inoculation		23.2%	51.3%	63%
Control		62.5%	96.7%	

* Calculated according to the formula in Section 2.5 and the survey data (*p* < 0.05).

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
