# Peer review of "Trichoderma harzianum Inoculation Reduces the Incidence of Clubroot Disease in Chinese Cabbage by Regulating the Rhizosphere Microbial Community"

_microorganisms, 2020, doi:10.3390/microorganisms8091325_

Round 1
Reviewer 1 Report
Following suggestions helps to improve this manuscript
- Many parts of the result section can be explained better with text rather than heavily depending on the figures.
- The discussion is quite weak and be improved by citing related work rather than just restating some of the results section
- Please provide better graphs where the names of the phyla or genera can be read (Figure 4).
- Authors need to follow the scientific nomenclature rule when mentioned bacterial and fungi names.
Reviewer 2 Report
This study deals with two objectives quantify (1) the effect of Trichoderma inoculation in the control of Plasmodiaphora brassicae clubroot disease, and (2) the additional impact of T. harzianum LTR-2 on rhizosphere microbial communities. Concerning to the first objective the data on Plasmodiaphora control is weakly presented. In table 2. It is showed the Disease Severity (raw data) no average data, no statistics and no data on disease incidence. The way the data is presented in Table 2 is not acceptable for a plant pathology study. It is claimed that the seed inoculation with strain LTR-2 of T. harzianum has an efficacy of 63% but the meaning of this figure depends on the levels of disease in the untreated plants. It is not the same the efficacy of 63% over a 80% disease incidence than an efficacy of 63% over a 20% of disease incidence. This should be clarified, and the plant disease data should be properly presented. This is important tin order to stablish the significance of the microbial data (objective 2).
Concerning to the second objective this manuscript presents a set of data showing the interactions of T harzianum with soil and rhizosphere microbiota in the context of Plasmodiaphora control. The sampling scheme of soil samples should be clarified, number of soil and rhizosphere samples per treatment…. The use of qPCR and high throughput sequencing should be justified. More references would be needed in the discussion and conclusion need to be more specific in relation to their findings.
In general, the manuscript should be thoroughly reviewed as many mistakes occur here and there. See specific comments.
My view is that the manuscript can be reconsidered for publication after major changes.
Specific comments
Abstract,
Sentence in line 4. This sentence is probably mistaken. I assume you think on T harzianum inoculation
Introduction
Line 5 Are these sinks located in galls, please specify
Material and methods
Sample collection. Remove the question mark ?
What is the purpose of using High-throughput sequencing?
What is the purpose of using High-throughput sequencing?
What does GSA stand for? Was this data processing applied to the high-throughput data?
Section 2.9. Why some words in this section are in red?
2nd paragraph. Please specify in what samples / factors are you going to show the differences.
Results
63% Where does this data come from?
Table 2. Not clear. Please remake it and clarify it.
Figure 1. Please specify the meaning of the abbreviations.
Section 3.2, Does this section refer only to fungi? Please clarify.
Trichoderma, Zopfiella, Orbiliaceae,… these are not species.
Figure 3. Is Rs the same as RsT, please unify notation
Figure 4. Legend difficult to read.
Figure 5. I am not familiar with Circos plot. I think many readers would need more information in order to be able to read it.
Discussion
Trichoderma harzianum should be in intalics.
Negative correlations between T harzianum and plant pathogens are not shown in figure 6. Did not find any figure 6b in the manuscript.
Section 4.2
…opposite to that of fungi. Any explanation on that?
Conclusions
Perhaps based on your results you can tell more details on relevant changes on soil and rhizosphere organisms.
Author Response
请参阅附件

Reviewer 3 Report
To the authors:
In the present study, titled ‘Trichoderma harzianum inoculation reduces the incidence of clubroot disease in Chinese cabbage by regulating the rhizosphere microbial community’, the authors examined the effect of Trichoderma harzianum application in Chinese cabbage under field conditions. The results showed that inoculation of T. harzianum greatly reduced in Plasmodiophora brassicae-infected soil the incidence and severity of cabbage clubroot in Chinese cabbage. The observed results coincided with significant changes to the microbial community in the rhizosphere and the bulk soil, most notably a significant reduction of plant pathogens, including P. brassicae. Indicating that T. harzianum can be an effective biological control agent for cabbage clubroot in field situations. The experimental design used in the manuscript was well structured to achieve the objectives presented. The results were clearly stated and will contribute to the scientific community interested in novel and sustainable management strategies.
As a reviewer, I appreciate the time and effort that went into the preparation of your article. Only a minor revision is needed to address the concerns/questions mentioned in the manuscript (pdf). After this revision, I would like to recommend its publication.

Round 2
Reviewer 1 Report
I think the English writing can be improved, it is very clear the newly added text in the discussion.
